# Cardiometabolic outcomes up to 12 months after COVID-19 infection. A matched cohort study in the UK

Emma Rezel-Potts[1,2], Abdel Douiri[1,2], Xiaohui Sun[1], Phillip J. Chowienczyk[2,3], Ajay M. Shah[2,3], Martin C. Gulliford[1,2]*

1 King's College London, School of Life Course & Population Sciences, London, United Kingdom, 2 National Institute for Health Research Biomedical Research Centre (BRC), Guy's and St Thomas' NHS Foundation Trust and King's College London, London, United Kingdom, 3 King's College London, British Heart Foundation Centre of Excellence, School of Cardiovascular Medicine & Sciences, London, United Kingdom

* martin.gulliford@kcl.ac.uk

## Abstract

### Background

Acute Coronavirus Disease 2019 (COVID-19) has been associated with new-onset cardiovascular disease (CVD) and diabetes mellitus (DM), but it is not known whether COVID-19 has long-term impacts on cardiometabolic outcomes. This study aimed to determine whether the incidence of new DM and CVDs are increased over 12 months after COVID-19 compared with matched controls.

### Methods and findings

We conducted a cohort study from 2020 to 2021 analysing electronic records for 1,356 United Kingdom family practices with a population of 13.4 million. Participants were 428,650 COVID-19 patients without DM or CVD who were individually matched with 428,650 control patients on age, sex, and family practice and followed up to January 2022. Outcomes were incidence of DM and CVD. A difference-in-difference analysis estimated the net effect of COVID-19 allowing for baseline differences, age, ethnicity, smoking, body mass index (BMI), systolic blood pressure, Charlson score, index month, and matched set. Follow-up time was divided into 4 weeks from index date ("acute COVID-19"), 5 to 12 weeks from index date ("post-acute COVID-19"), and 13 to 52 weeks from index date ("long COVID-19"). Net incidence of DM increased in the first 4 weeks after COVID-19 (adjusted rate ratio, RR 1.81, 95% confidence interval (CI) 1.51 to 2.19) and remained elevated from 5 to 12 weeks (RR 1.27, 1.11 to 1.46) but not from 13 to 52 weeks overall (1.07, 0.99 to 1.16). Acute COVID-19 was associated with net increased CVD incidence (5.82, 4.82 to 7.03) including pulmonary embolism (RR 11.51, 7.07 to 18.73), atrial arrythmias (6.44, 4.17 to 9.96), and venous thromboses (5.43, 3.27 to 9.01). CVD incidence declined from 5 to 12 weeks (RR 1.49, 1.28 to 1.73) and showed a net decrease from 13 to 52 weeks (0.80, 0.73 to 0.88). The analyses were based on health records data and participants' exposure and outcome status might have been misclassified.

**Data Availability Statement:** The study is based on data from the Clinical Practice Research Datalink (CPRD) obtained under license from the UK Medicines and Healthcare Products Regulatory

Agency (MHRA); however, the interpretation and conclusions contained in this report are those of the authors alone. Requests for access to data from the study should be addressed to cprdenquiries@mhra.gov.uk. All proposals requesting data access will require approval from CPRD before data release.

**Funding:** ERP, AD, PJC, AMS and MG acknowledge support by the NIHR Biomedical Research Centre at Guy's and St Thomas' NHS Foundation Trust in partnership with King's College London (IS-BRC-1215-20006). AMS is also supported by the British Heart Foundation (RE/18/2/34213). The funder of the study had no role in study design, data collection, data analysis, data interpretation, or writing of the report.

**Competing interests:** The authors have declared that no competing interests exist.

**Abbreviations:** BMI, body mass index; CI, confidence interval; COVID-19, Coronavirus Disease 2019; CPRD, Clinical Practice Research Datalink; CVD, cardiovascular disease; DBP, diastolic blood pressure; DM, diabetes mellitus; IL-6, interleukin-6; PCR, polymerase chain reaction; SARS-CoV-2, Severe Acute Respiratory Syndrome Coronavirus 2; SBP, systolic blood pressure; TNFα, tumour necrosis factor alpha.

## Conclusions

In this study, we found that CVD was increased early after COVID-19 mainly from pulmonary embolism, atrial arrhythmias, and venous thromboses. DM incidence remained elevated for at least 12 weeks following COVID-19 before declining. People without preexisting CVD or DM who suffer from COVID-19 do not appear to have a long-term increase in incidence of these conditions.

## Author summary

### Why was this study done?

➤ Acute Coronavirus Disease 2019 (COVID-19) may be associated with cardiovascular complications and disorders of blood glucose.

➤ It is not known whether patients recovering from COVID-19 remain at increased risk of cardiovascular disease (CVD) or diabetes mellitus (DM).

➤ This study aimed to find out whether new diagnoses of DM and CVDs are increased over 12 months after COVID-19 compared with matched control patients who did not have COVID-19.

### What did the researchers do and find?

➤ We analysed electronic records for 428,650 COVID-19 patients who were matched with 428,650 control patients and followed up to January 2022. We evaluated new diagnoses of DM and CVD up to 12 months after COVID-19 infection. We compared COVID-19 patients with controls and adjusted for baseline differences in risk.

➤ DM diagnoses were increased by 81% in acute COVID-19 and remained elevated by 27% from 4 to 12 weeks after the infection.

➤ Acute COVID-19 was associated with a 6-fold increase in cardiovascular diagnoses overall, including an 11-fold increase in pulmonary embolism, a 6-fold increase in atrial arrythmias, and a 5-fold increase in venous thromboses. CVD diagnoses declined from 4 to 12 weeks after COVID-19 and returned to baseline levels or below from 12 weeks to 1 year after the infection.

### What do these findings mean?

➤ Acute COVID-19 is associated with increased risk of cardiovascular disorders, but risk generally returns to background levels soon after the infection.

➤ The risk of new DM remains increased for at least 12 weeks following COVID-19 before declining.

> ➤ Patients recovering from COVID-19 should be advised to consider measures to reduce diabetes risk including healthy diet and taking exercise.

> ➤ People without preexisting CVD or DM who suffer from COVID-19 do not appear to have a long-term increase in incidence of these conditions.

## Introduction

Coronavirus Disease 2019 (COVID-19) is increasingly recognised as a multisystem condition [1]. Infection of the respiratory tract by the Severe Acute Respiratory Syndrome Coronavirus 2 (SARS-CoV-2) virus triggers host immune responses that may have systemic effects through activation of inflammatory pathways [2,3]. COVID-19 may trigger a proinflammatory "cyto-kine storm" with dysregulated immune response, platelet activation, hypercoagulability, endothelial cell dysfunction, and thromboembolism affecting diverse systems with potential for end-organ damage [4]. While acute COVID-19 infection has been associated with new-onset cardiovascular disease (CVD) and diabetes mellitus (DM) [5], longer-term outcomes up to 1 year following the infection have not been well characterised. Cardiac manifestations of COVID-19 [6,7] include cardiac injury with elevated troponin levels, heart failure, and increased risk of mortality among patients hospitalised with COVID-19 [8,9]. COVID-19 may also be associated with acute myocardial infarction and ischaemic stroke in the first 4 weeks [10]. New-onset hyperglycaemia has been reported in COVID-19 patients, sometimes representing "stress hyperglycaemia," and is associated with worse prognosis [5,11]. Complications of both preexisting and new-onset DM have been observed, including diabetic ketoacidosis and hyperosmolarity [12–14]. Possible mechanisms appear to be direct pancreatic damage from SARS-CoV-2 and/or the associated systemic inflammatory syndrome with severe COVID-19, as reflected in high levels of interleukin-6 (IL-6) and tumour necrosis factor alpha (TNFα), causing impaired pancreatic insulin secretion [15] and insulin resistance [16].

Since the first evidence of COVID-19 emerged in the beginning of 2020, there have been multiple waves of infection and multiple variants of SARS-CoV-2 have been identified. There were in excess of 1,000 deaths per day in the United Kingdom at the peak of the first wave in April 2020 and the peak of the second wave in January 2021 with many more suffering illness of varying severity [17]. The longer-term outcomes of COVID-19 are now receiving increasing attention. A high proportion of patients report experiencing symptoms for more than 4 weeks after initial presentation with COVID-19 [18]. The distinction has been made between "acute COVID-19" in the first 4 weeks after infection; "post-acute COVID-19" (or "ongoing symptomatic COVID-19"), from 5 to 12 weeks after the first infection; and "long COVID-19" (or "post-COVID-19 syndrome") with symptoms persisting for more than 12 weeks after infection [19]. There is concern that cardiovascular and metabolic outcomes may be compromised in the aftermath of COVID-19 infection. However, susceptibility to COVID-19 and the severity of illness are known to be associated with cardiometabolic risk. Furthermore, public health restrictions during the height of the pandemic, including "lockdowns" or "stay-at-home" orders, were associated with profound changes in diet, exercise habits, and other health-related behaviours that might have impacted on CVD and diabetes in the general population even in the absence of COVID-19 infection. Controlled studies are therefore needed to evaluate the net long-term impacts of COVID-19 infection on cardiovascular and diabetes outcomes after allowing for premorbid differences between cases and controls and changes over

time in control participants. There is concern for the possible burden of "long COVID-19" syndromes in patients recovering from COVID-19, but few studies have reported on long-term follow-up for large population-based samples. Al-Aly and colleagues [5] employed a data mining approach to Veterans' Administration data and identified an increased burden of diverse health conditions between 30 days and 6 months after COVID-19. Knight and colleagues [20] suggested that arterial and venous complications remain elevated for 49 weeks after COVID-19. However, the timeline for recovery from COVID-19 remains poorly characterised.

Longitudinal datasets from electronic health records offer opportunities to analyse longer-term COVID-19 outcomes. We employed the Clinical Practice Research Datalink (CPRD), a database of anonymised primary care electronic health records, to identify a cohort of COVID-19 exposed patients in comparison with a matched cohort with no COVID-19 diagnosis. We aimed to estimate the net effect of COVID-19 on cardiometabolic outcomes over periods of 4 weeks, 3 months, and 12 months in order to inform research priorities, clinical services, and public health interventions that may be required following acute COVID-19 infection.

## Methods

### Ethical approval

The protocol was given scientific and ethical approval by the CPRD Independent Scientific Advisory Committee (ISAC protocol 20_00265R). CPRD holds over-arching research ethics committee approval for the conduct of research using fully anonymised electronic health records data from the CPRD databases. The study protocol has been published in summary form by the CPRD/MHRA and can be accessed here. The full protocol, including the prospective analysis plan, is included as S1 Protocol. This study is reported as per the Strengthening the Reporting of Observational Studies in Epidemiology (STROBE) guideline (S1 Checklist).

### Data source and participant selection

We conducted a population-based, matched cohort study in CPRD Aurum. The CPRD Aurum is a large database that includes comprehensive medical record data for a current total of 1,356 family practices in England with approximately 13.4 million currently registered patients in the March 2022 release [21]. The database includes data for all registered patients at participating general practices, except for a negligible number of patients who opt out of data collection. Each patient has a unique anonymised numerical identifier that remains the same at each update of the database. Patients may therefore be tracked through successive releases of the database which is updated monthly. The March 2022 release data, which cover approximately 20% of the population of England, are considered to be generally representative of the general population with regards to geographical distribution, deprivation, age, and sex [21]. COVID-19 test results are automatically transmitted from laboratories to patients' family practices. Coded records for CVD or DM derive from general practice consultations, hospital outpatient referrals, and inpatient admissions.

At the inception of the study, data for all patients with a diagnosis of COVID-19 were extracted from the February 2021 release of CPRD Aurum. The date of the first code for COVID-19 was the index date. Since confirmatory testing was not widely available during the initial phase of the pandemic, we included patients with clinical diagnoses of "confirmed" or "suspected" COVID-19 (S1 Table). However, we conducted a sensitivity analysis including only patients with a medical code recorded for polymerase chain reaction (PCR) test confirmed COVID-19. The COVID-19 cohort was compared with a sample of matched control

patients who were not recorded with a COVID-19 diagnosis up to the case index date. Control patients were randomly sampled from the registered population of CPRD Aurum March 2021 release that provided the most up-to-date information on the database at the time of sampling. Controls were matched for year of birth, sex, and family practice, and their record was required to start no later than 18 months after the start of record for their matched case. Patients were excluded from eligibility as controls if their record ended before the matched case index date, or if they had prevalent CVD or DM recorded more than 12 months before the index date or within 1 year of the start of their record. Using the unique identifiers for patients in the sample, data were updated to the March 2022 release of CPRD Aurum for final analyses, providing follow-up to 31 January 2022. The records of any control patients diagnosed with COVID-19 after the index date were censored 30 days before the control COVID-19 diagnosis date (S1 Text).

## Outcome measures

The study outcomes were first ever recorded CVD and DM diagnoses (S1 Table). CVD diagnoses were grouped into the categories: myocardial infarction and ischaemic heart disease; atrial arrhythmias, including atrial fibrillation and supraventricular tachycardia; heart failure; cardiomyopathy and myocarditis; pulmonary embolism; venous thrombosis; and stroke. DM diagnoses included diagnoses for type 1 and type 2 DM and initiation of oral hypoglycaemic drugs and insulin. Records of HbA1c were evaluated, and a second record of HbA1c $\geq$48 mmol/mol was considered diagnostic of diabetes. Patients were considered to have a type 1 DM phenotype if they were aged 35 years or less at diagnosis and were prescribed insulin within 91 days of diagnosis [22]. Mortality was evaluated from the CPRD date of death.

## Covariates

Covariates were defined using data recorded in the study period before the index date. Covariates selected because of known associations with CVD and DM included smoking status (nonsmoker, current smoker, ex-smoker), body mass index (BMI, underweight <18.5 kg/m$^2$, normal weight 18.5 to <25.0 kg/m$^2$, overweight 25.0 to <30.0 kg/m$^2$, obese $\geq$30 kg/m$^2$, and not recorded), systolic blood pressure (SBP), and diastolic blood pressure (DBP) in categories of 10 mm Hg. Ethnicity was classified as "white," "black," "Asian" (of Indian subcontinent origins), "mixed," "other," and "not known." We also employed the Charlson index to summarize comorbidities that may be associated with the severity of illness in COVID-19 [23]. Each component morbidity was classified as present or absent before the index date; the Charlson score was calculated with a maximum of 4 or more [24,25]. COVID-19 and control patients were matched on sex, year of birth, and family practice, with the latter offering control for area measures including level of deprivation. Prescriptions for glucocorticoids were evaluated in patients diagnosed with DM because this therapy for severe COVID-19 is associated with diabetes risk [26].

## Analysis

The prospective plan envisaged analysis in a time-to-event framework, with the date of COVID-19 as the start date (S1 Protocol). Recognition of the importance of comparing risks from before the index date and adjusting for baseline values in a difference-in-difference analysis prompted us to employ Poisson models to evaluate incidence of DM and CVD in periods before and after the COVID-19 diagnosis. However, time-to-event and log-linear models are generally equivalent and give similar results [27].

Person-time at risk was evaluated in terms of person-weeks of follow-up, and incident events of CVD and DM were identified in each 4-week period. Follow-up time was divided

into periods of 28 days, from 1 year before the index date to 1 year after the index date. Data were then aggregated for the periods before the index date ("Pre-Index") and for 4 weeks after the index date ("acute COVID-19"), 5 to 12 weeks after the index date ("Post-acute COVID-19") and from 13 to 52 weeks after the index date ("Long COVID-19"). Incidence rates, with exact Poisson confidence intervals (CIs), were estimated for COVID-19 patients and controls. Poisson regression models were fitted using the method of generalised estimating equations, employing an exchangeable correlation structure with robust standard error estimates (S2 Text). This approach allowed for clustering on the matched set of COVID-19 participant and control, as well as allowing for possible overdispersion. The effect of group (COVID-19 or control), time (pre-index, 4 weeks from index, 5 to 12 weeks, and 13 to 52 weeks), and the group-time interaction were estimated. The group-time interaction represented the net additional effect of acute COVID-19, post-acute COVID-19, or long COVID-19 in COVID-19 participants, net of the baseline values for COVID-19 participants, and the comparison with control participants. Analyses were adjusted for age, age-squared, sex, ethnicity, BMI, SBP, Charlson score, index month, and index month squared (S2 Text). Missing values were represented by indicator variables (Table 1); we compared unadjusted and covariate adjusted estimates. We recognised that incidence of CVD and DM might change during the period from 13 to 52 weeks after COVID-19 diagnosis. Therefore, in secondary analyses, we estimated adjusted rate ratios and their CIs for the comparison of each 4-week period following COVID-19 diagnosis with baseline, using the same analytical approach. Estimates were plotted with a smoothed curve fitted using the loess method. We conducted a sensitivity analysis including only COVID-19 participants with a positive PCR test for SARS-CoV-19 infection. A logistic regression model was fitted to evaluate variables associated with PCR confirmation. All analyses were implemented in the R program, version 3.6.1. Following peer review, we included a sensitivity analysis to evaluate whether adjusting for consultation frequency might explain association of COVID-19 with diabetes incidence.

## Results

There were 516,985 participants diagnosed with COVID-19. COVID-19 participants with prevalent CVD or DM, diagnosed within 1 year of the start of record or more than 1 year before the index date, were excluded leaving 431,193 COVID-19 participants. These were matched with potential controls and after excluding participants with indeterminate sex, age more than 104 years, prevalent CVD or DM recorded in their full medical record, there were 428,650 (99%) matched sets, with a median duration of follow-up of 12 months, for further analysis (Table 1 and S1 Text). COVID-19 participants included 347,011 (80.9%) with clinical or laboratory confirmation of diagnosis on the index date and 281,486 (65.7%) with a positive PCR test for SARS-CoV-2 on the index date (S3 Text). COVID-19 participants and controls were similar with respect to matching variables of age and sex, there was a slight female predominance with median age of 35 years (Table 1). COVID-19 participants were more likely to have defined values recorded for covariates. COVID-19 participants included slightly more of "Asian" ethnicity, fewer current smokers, more who were overweight and obese, and more with comorbidity than controls, while blood pressure distributions appeared to be similar. COVID-19 participants had a median of 11 consultations (interquartile range 6 to 21) after the index date, compared with 7 (3 to 15) in controls.

Fig 1 shows the incidence of DM and CVD from 52 weeks before the index date to 52 weeks after for COVID-19 participants (red) and controls (blue). In the period before the index date, the incidence of diabetes was slightly higher in patients who went on to be diagnosed with COVID-19 than control participants; a similar difference was also observed for the

**Table 1. Characteristics of COVID-19 and matched control participants.** Figures are frequencies (percent of column total).

| | | Matched controls (428,650) | COVID-19 patients (428,650) |
|---|---|---|---|
| **Sex** | Male | 190,401 (44) | 190,401 (44) |
| | Female | 238,249 (56) | 238,249 (56) |
| **Age (median, IQR years)** | | 35 (22 to 50) | 35 (22 to 50) |
| **Ethnicity** | "Asian" | 25,978 (6) | 38,164 (9) |
| | "Black African/Caribbean" | 15,747 (4) | 15,783 (4) |
| | "Mixed" | 7,285 (2) | 9,278 (2) |
| | Not recorded | 130,560 (30) | 83,850 (20) |
| | "Other" | 14,760 (3) | 17,205 (4) |
| | "White" | 234,320 (55) | 264,370 62) |
| **Cigarette smoking** | Nonsmoker | 201,549 (47) | 226,339 (53) |
| | Ex-smoker | 51,333 (12) | 64,320 (15) |
| | Current smoker | 79,285 (18) | 67,979 (16) |
| | Not recorded | 96,483 (23) | 70,012 (16) |
| **BMI** | Underweight | 14,596 (3) | 16,432 (4) |
| | Normal weight | 89,903 (21) | 102,241 (24) |
| | Overweight | 66,400 (15) | 84,393 (20) |
| | Obese | 51,695 (12) | 70,854 (17) |
| | Not recorded | 206,056 (48) | 154,730 (36) |
| **SBP** | <110 | 45,989 (11) | 52,133 (12) |
| | 110–119 | 67,188 (16) | 74,917 (17) |
| | 120–129 | 82,021 (19) | 91,718 (21) |
| | 130–139 | 64,381 (15) | 70035 (16) |
| | 140–149 | 28,642 (7) | 29,874 (7) |
| | 150–159 | 8,058 (2) | 8,503 (2) |
| | ≥160 | 5,067 (1) | 5,307 (1) |
| | Not recorded | 127,304 (30) | 96,163 (22) |
| **Charlson score** | 0 | 341,996 (80) | 330,139 (77) |
| | 1 | 70,581 (16) | 80,337 (19) |
| | 2 | 11,438 (3) | 12,127 (3) |
| | 3 | 3,197 (1) | 4,141 (1) |
| | ≥4 | 1,438 (0) | 1,906 (0) |

BMI, body mass index; COVID-19, Coronavirus Disease 2019; IQR, interquartile range; SBP, systolic blood pressure.

incidence of CVDs. In the first 4 weeks after COVID-19 diagnosis, there was a sharp increase in the incidence of diabetes and an even greater increase in CVDs. In the remainder of the first year following COVID-19 diagnosis, the incidence of DM for COVID-19 cases appeared to remain higher than for controls, while CVD incidence appeared to decline to premorbid values.

Table 2 shows the incidence of DM and CVD aggregated into 4 periods: before the index date, within 4 weeks of the index date, from 5 to 12 weeks after the index date, and from 13 to 52 weeks after the index date. In the period before the index date, the incidence of diabetes was 15.81 (95% CI 15.29 to 16.34) per 100,000 patient weeks in patients who were later diagnosed with COVID-19 but 11.32 (10.88 to 11.77) in controls who did not develop COVID-19; the incidence of CVD was 14.07 (13.58 to 14.58) per 100,000 patient weeks for cases and 7.58 (7.22 to 7.95) for controls, respectively. During the first 4 weeks from COVID-19 diagnosis, the incidence of DM increased to 23.79 (21.57 to 26.18) and the incidence of CVD increased to 76.92

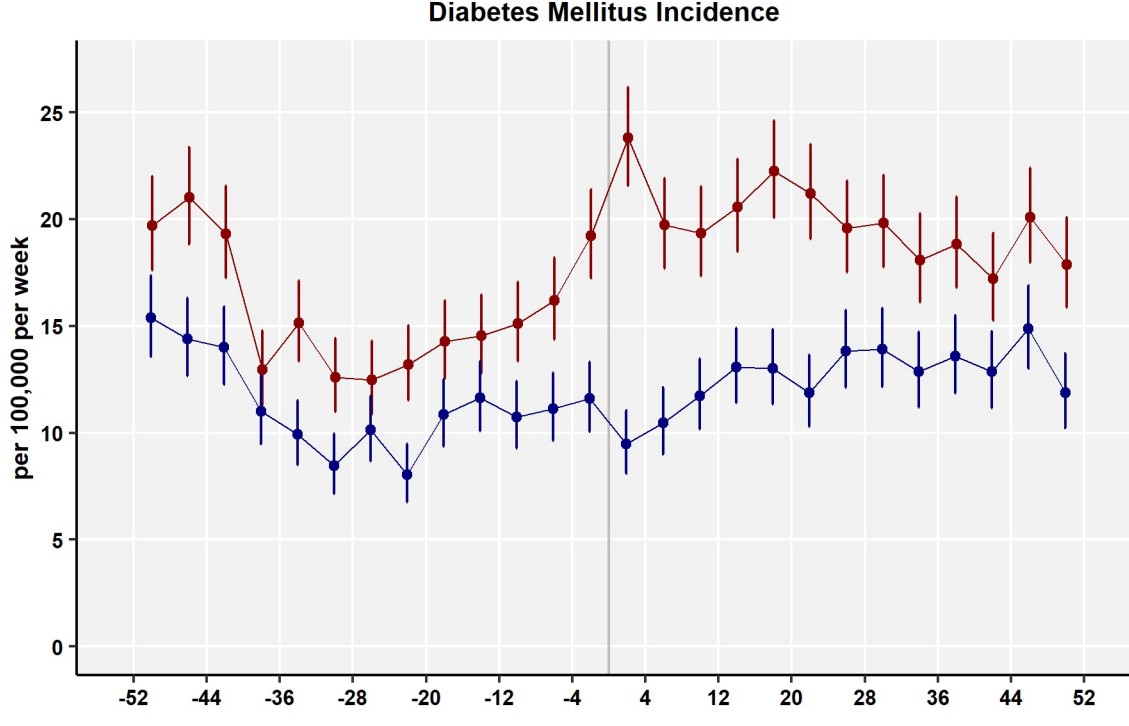

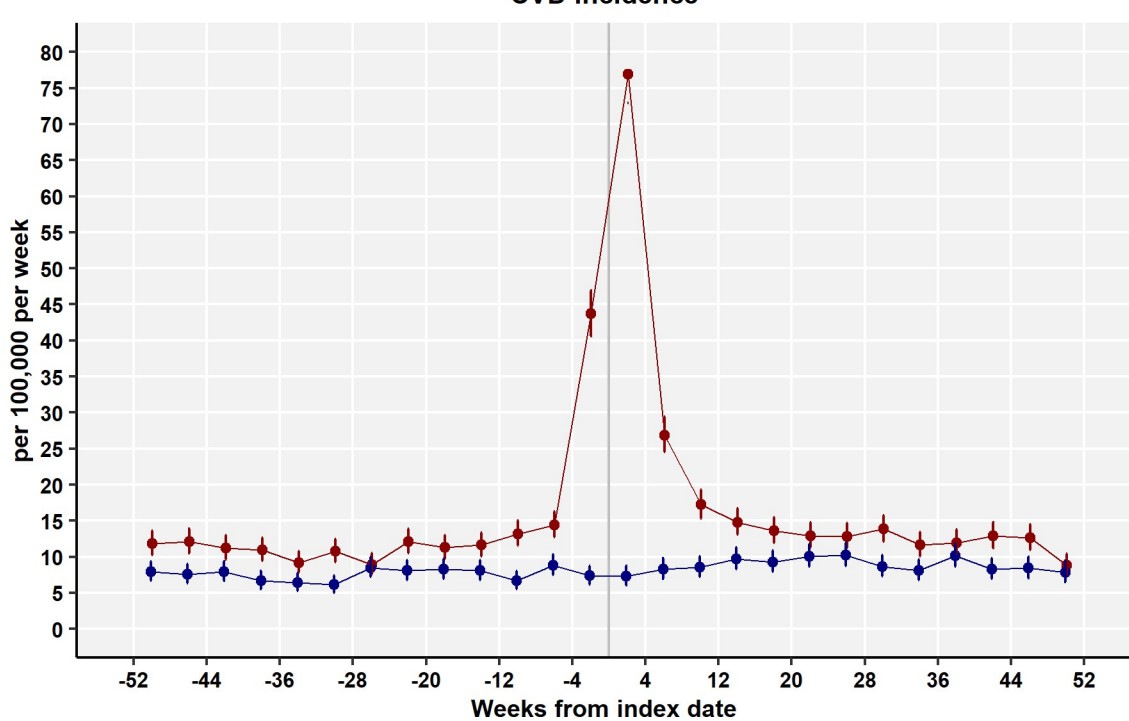

**Fig 1. Incidence rates for DM and CVDs (per 100,000 patient weeks) for COVID-19 patients (red) and controls (blue) for 4-week periods.** Bars are 95% CIs. CI, confidence interval; COVID-19, Coronavirus Disease 2019; CVD, cardiovascular disease; DM, diabetes mellitus.

**Table 2. DM and CVD events and rates by period for COVID-19 cases and controls.** Figures are frequencies except where indicated.

| Phase | COVID-19 or control | Patient weeks | CVD events | CVD incidence per 100,000 patient weeks (95% CI) | Diabetes diagnoses | DM incidence per 100,000 patient weeks (95% CI) |
|---|---|---|---|---|---|---|
| Before index date | COVID-19 | 21,894,712 | 3,081 | 14.07 (13.58 to 14.58) | 3,461 | 15.81 (15.29 to 16.34) |
| | Controls | 22,462,441 | 1,702 | 7.58 (7.22 to 7.95) | 2,542 | 11.32 (10.88 to 11.77) |
| Acute: up to 4 weeks from index | COVID-19 | 1,765,392 | 1,358 | 76.92 (72.89 to 81.13) | 420 | 23.79 (21.57 to 26.18) |
| | Controls | 1,750,425 | 128 | 7.31 (6.10 to 8.69) | 166 | 9.48 (8.10 to 11.04) |
| Post-acute: 5 to 12 weeks from index | COVID-19 | 3,485,790 | 769 | 22.06 (20.53 to 23.68) | 681 | 19.54 (18.10 to 21.06) |
| | Controls | 3,461,014 | 291 | 8.41 (7.47 to 9.43) | 384 | 11.10 (10.01 to 12.26) |
| Long: 13 to 52 weeks from index | COVID-19 | 16,635,214 | 2,101 | 12.63 (12.10 to 13.18) | 3,256 | 19.57 (18.91 to 20.26) |
| | Controls | 16,351,115 | 1,487 | 9.09 (8.64 to 9.57) | 2,153 | 13.17 (12.62 to 13.74) |

CI, confidence interval; COVID-19, Coronavirus Disease 2019; CVD, cardiovascular disease; DM, diabetes mellitus.

(72.89 to 81.13) per 100,000 patient weeks. In the periods from 5 to 12 weeks and 13 to 52 weeks, the incidence of CVD declined to 22.06 (20.53 to 23.68) and 12.63 (12.10 to 13.18) per 100,000 patient weeks, respectively. Thus, in the period from 13 to 52 weeks after COVID-19 diagnosis, the overall incidence of CVD declined to premorbid levels or below. During the same periods, the incidence of DM appeared to remain elevated at 19.54 (18.10 to 21.06) per 100,000 in the post-acute period and 19.57 (18.91 to 20.26) per 100,000 in the long-COVID period.

Fig 2 shows the incidence of categories of CVD from 52 weeks before the COVID-19 index date to 52 weeks after. Pulmonary embolism showed the greatest increase during the acute COVID-19 period, with 34.67 cases per 100,000 patient weeks. The value was higher than for atrial arrhythmias (13.93 per 100,000 per week), venous thrombosis (8.84 per 100,000 per week), myocardial infarction and ischaemic heart disease (6.57 per 100,000 per week), stroke (6.68 per 100,000 per week), heart failure (3.80 per 100,000 per week), or cardiomyopathy and myocarditis (2.21 per 100,000 per week). While the relative effects differed for categories of CVD, the time course of changes in incidence was similar for each category of CVD.

Table 3 presents estimated incidence rate ratios both unadjusted and adjusted for age, sex, ethnicity, smoking, BMI, SBP, and Charlson comorbidity score. After allowing for underlying differences between cases and controls, and between the baseline pre-index period and follow-up, the net effect of acute COVID-19 on CVD incidence was estimated to be an adjusted rate ratio of 5.82 (95% CI 4.82 to 7.03), while for DM incidence the adjusted rate ratio was 1.81 (1.51 to 2.19). In the period from 5 to 12 weeks after diagnosis, the adjusted incidence rate ratio for CVD declined to 1.49 (1.28 to 1.73), while for DM, this was 1.27 (1.11 to 1.46) (Table 3 and S4 Text). In the period from 13 to 52 weeks after diagnosis, the adjusted incidence rate ratio for CVD was 0.80 (0.73 to 0.88), while for DM, this was 1.07 (0.99 to 1.16). Covariate adjustment resulted in little change in the estimated incident rate ratios (Table 3), with the difference-in-difference analysis already allowing for baseline differences in risk.

When categories of CVD were analysed separately (Table 3), pulmonary embolism (RR 11.51, 7.07 to 18.73), atrial arrhythmias (6.44, 4.17 to 9.96), and venous thromboses (5.43, 3.27 to 9.01) showed the greatest relative increases in acute COVID-19 but myocardial infarction (2.01, 1.34 to 3.00), heart failure (5.23, 2.04 to 13.44), and stroke (3.31, 2.05 to 5.35) were also increased. These conditions declined substantially in the period from 5 to 12 weeks after COVID-19 diagnosis. There was no evidence that any of these categories of CVD showed a net increase during the period from 13 to 52 weeks after COVID-19 diagnosis, with evidence of lower incidence for several, including pulmonary embolism and stroke.

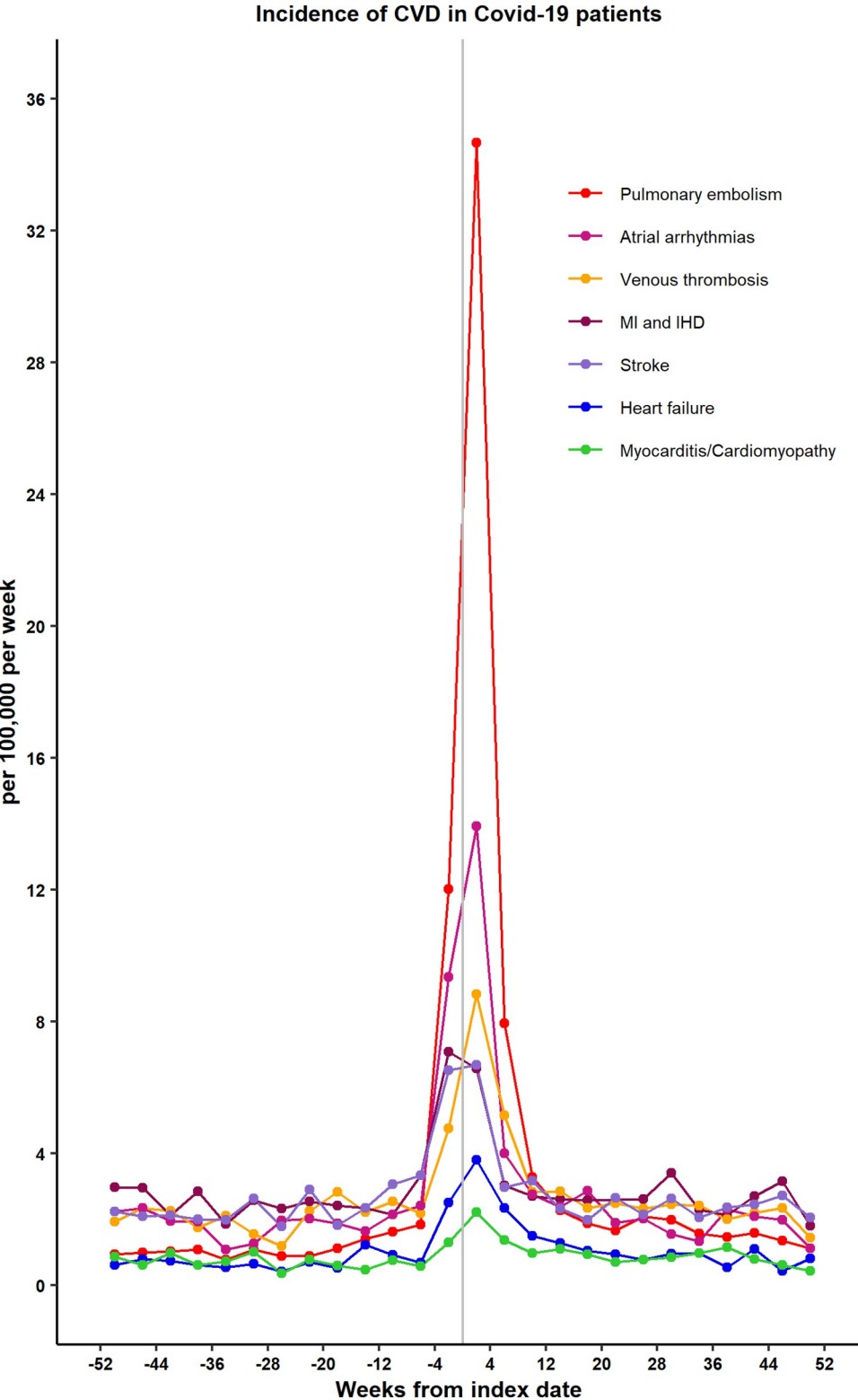

**Fig 2. Incidence rates for categories of CVD (per 100,000 patient weeks) for COVID-19 patients by 4-week periods.** COVID-19, Coronavirus Disease 2019; CVD, cardiovascular disease; IHD, ischaemic heart disease; MI, myocardial infarction.

**Table 3. Results of difference-in-difference analysis showing net effect of COVID-19 over baseline rate and comparison with controls.** (Estimates were adjusted for age, ethnicity, smoking, BMI category, SBP category, Charlson score, index month, and matched set).

| | | "Acute COVID-19" up to 4 weeks | | "Post-acute COVID-19" 5 to 12 weeks | | "Long COVID-19" 13 to 52 weeks | |
|---|---|---|---|---|---|---|---|
| | | RR (95% CI) | *P* value | RR (95% CI) | *P* value | RR (95% CI) | *P* value |
| **DM** | **Unadjusted** | 1.80 (1.49 to 2.17) | <0.001 | 1.26 (1.10 to 1.45) | <0.001 | 1.07 (0.99 to 1.15) | 0.09 |
| | **Adjusted** | 1.81 (1.51 to 2.19) | <0.001 | 1.27 (1.11 to 1.46) | <0.001 | 1.07 (0.99 to 1.16) | 0.07 |
| **All CVD outcomes** | **Unadjusted** | 5.68 (4.70 to 6.86) | <0.001 | 1.42 (1.22 to 1.65) | <0.001 | 0.76 (0.69 to 0.83) | <0.001 |
| | **Adjusted** | 5.82 (4.82 to 7.03) | <0.001 | 1.49 (1.28 to 1.73) | <0.001 | 0.80 (0.73 to 0.88) | <0.001 |
| **Subgroups of CVD. Adjusted estimates** | | | | | | | |
| Pulmonary embolism | | 11.51 (7.07 to 18.73) | <0.001 | 2.29 (1.47 to 3.57) | <0.001 | 0.66 (0.50 to 0.87) | 0.003 |
| Atrial arrhythmias | | 6.44 (4.17 to 9.96) | <0.001 | 1.58 (1.10 to 2.27) | 0.01 | 0.85 (0.68 to 1.05) | 0.13 |
| Venous thrombosis | | 5.43 (3.27 to 9.01) | <0.001 | 1.84 (1.27 to 2.64) | 0.001 | 0.88 (0.71 to 1.10) | 0.26 |
| Heart failure | | 5.23 (2.04 to 13.44) | <0.001 | 2.30 (1.17 to 4.51) | 0.02 | 0.72 (0.49 to 1.06) | 0.10 |
| Stroke | | 3.31 (2.05 to 5.35) | <0.001 | 0.87 (0.63 to 1.21) | 0.41 | 0.73 (0.59 to 0.89) | 0.002 |
| Cardiomyopathy and myocarditis | | 2.96 (1.27 to 6.88) | 0.01 | 1.37 (0.70 to 2.68) | 0.35 | 1.11 (0.74 to 1.66) | 0.62 |
| Myocardial infarction and IHD | | 2.01 (1.34 to 3.00) | <0.001 | 0.97 (0.68 to 1.38) | 0.86 | 0.80 (0.66 to 0.98) | 0.03 |

BMI, body mass index; CI, confidence interval; COVID-19, Coronavirus Disease 2019; CVD, cardiovascular disease; DM, diabetes mellitus; IHD, ischaemic heart disease; RR, adjusted incidence rate ratio; SBP, systolic blood pressure.

When adjusted rate ratios were estimated by 4-week period (Fig 3 and S5 Text), there was no evidence that CVD events might be increased beyond 8 to 11 weeks after COVID-19 onset (RR 1.15, 95% CI 0.94 to 1.42, *P* = 0.18). There was evidence of a net reduction in CVD events, compared with baseline and controls, from weeks 20 to 23 onwards. There was evidence that DM incidence might be increased up to 20 to 23 weeks after COVID-19 onset (1.29, 1.08 to 1.55, *P* = 0.005) but not from 24 to 27 weeks onwards (1.03, 0.86 to 1.22, *P* = 0.78).

The median age at diagnosis of DM was 47 (interquartile range 34 to 57) years for COVID-19 cases and 44 (33 to 57) years for controls; for CVD, the median age at diagnosis was 60 (49 to 73) for COVID-19 cases and 60 (49 to 72) for controls (S6 Text). The distribution of sex, ethnicity, smoking, and obesity were generally consistent across stages of COVID-19 infection. Patients diagnosed with diabetes in the context of acute COVID-19 were more likely to be prescribed insulin with 91 days of diagnosis (52/420, 12%, compared with 11/166, 7%, for controls) but there was no clear evidence of an increase in type 1 DM at any stage of the illness (S6 Text). There were 7,724 (1.8%) control patients and 18,895 (4.4%) COVID-19 patients who received one or more primary care prescriptions for oral, enteral, intramuscular, or intravenous glucocorticoids from 4 weeks before the index date to the end of follow-up. There were 126 new diagnoses of diabetes in glucocorticoid-treated controls and 417 in glucocorticoid-treated cases during the same period.

## Sensitivity analysis

There were 281,486 (65.7%) COVID-19 participants with a positive PCR test recorded on the index date. Compared with the entire sample, patients with PCR test confirmation of COVID-19 tended to be younger, were more often white, male, nonobese, and nonsmokers. The odds of PCR confirmation decreased by 1.6% per year increase in age. Results of the sensitivity analysis are shown in S7 Text. Although rates of CVD and DM were lower in the sample with PCR confirmed COVID-19 infection, consistent with their generally lower cardiometabolic risk

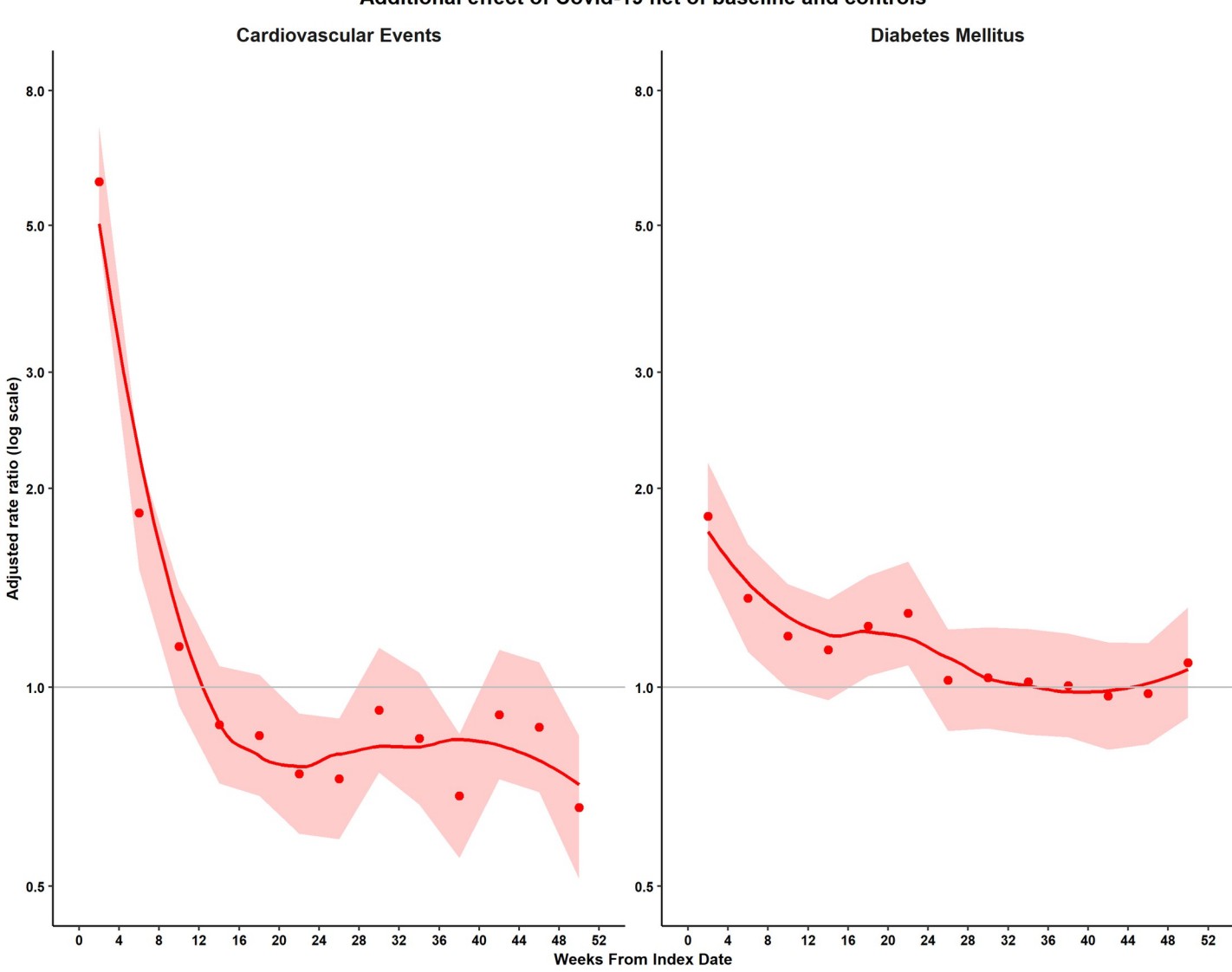

**Fig 3. Adjusted rate ratios by 4-week period following COVID-19.** Points represent adjusted rate ratios, bands represent 95% CIs, line represents smoothed curve. (Estimates were adjusted for age, ethnicity, smoking, BMI category, SBP category, Charlson score, index month, and matched set.) BMI, body mass index; CI, confidence interval; COVID-19, Coronavirus Disease 2019; SBP, systolic blood pressure.

profile, differences between COVID-19 cases and controls, and changes over time following COVID-19 infection, were consistent with those observed in the whole sample. In the analysis for DM, adjustment for consultation frequency had negligible effect on the first 2 decimal places of regression coefficients and their standard errors (S8 Text).

## Discussion

### Main findings

This study compared COVID-19 patients with matched population controls from 1 year before a COVID-19 diagnosis to 1 year after. The study showed that patients recorded with COVID-19 in primary care are at slightly greater risk of CVD and DM even before the

COVID-19 diagnosis. While patients with preexisting DM or CVD may be at greater risk of symptomatic COVID-19, we excluded all patients with DM or CVD as preexisting conditions at study entry. Patients who had already died from CVD or DM were also necessarily excluded. Elevated cardiometabolic risk was associated with higher BMI, greater comorbidity, and a higher proportion of "Asian" ethnicity. After allowing for baseline differences between COVID-19 patients and controls, new DM diagnoses increased 1.8-fold during acute COVID-19 and continued to show evidence of increase for at least 12 weeks before declining. There was no evidence overall that COVID-19 was associated with any additional increase in DM and CVD events from 13 to 52 weeks, above that observed in controls. However, secondary analysis by 4-week period provided evidence that DM events might not return to baseline until 20 to 23 weeks after COVID-19. There was evidence of increased glucocorticoid exposure in COVID-19 patients, but this might be associated with only a very small proportion of new DM events. New CVD events were increased nearly 6-fold during acute COVID-19 and nearly 50% in post-acute COVID-19. This increase was largely driven by pulmonary embolism diagnoses, but atrial arrythmias, venous thrombosis, myocardial infarction, stroke, and heart failure also showed more modest increases. Cardiovascular conditions are often associated with acute presentations that may lead to early diagnosis; this is in contrast to diabetes which may remain undiagnosed for a variable period of time, possibly contributing to the more delayed decline in the latter condition with consultation rates changing over time [28].

## Comparison with other studies

The long-term outcomes of COVID-19 infection represent an emerging area of research and public concern, but definitions are not universally agreed. This study drew on clinical recommendations that distinguish "acute" (4 weeks), "post-acute" (3 months), and "long" (1 year) periods after COVID-19 infection [19]. "Long COVID" is often self-identified by patients with the concept focusing on symptom burden and impacts on health-related quality of life [19,29]. These subjective outcomes of COVID-19 infection may be heterogenous in nature and their occurrence and persistence may depend on patient characteristics including age, sex, and comorbidity, as well as the severity of the initial COVID-19 illness and intensity and duration of therapeutic support in the acute illness [29,30]. It is well established that preexisting hypertension and ischaemic heart disease are associated with more severe illness and greater mortality in acute COVID-19 infection [31]. Hospitalised patients may experience a range of cardiovascular complications including arrhythmias, heart failure, and thrombotic complications [31], but there are few studies with long-term follow-up of patients without preexisting CVD. In a preprint, Knight and colleagues [20] reported on cardiovascular outcomes of a large population in England. Their analyses suggested that CVD outcomes might remain increased for up to 49 weeks following COVID-19 infection [20]. However, the analyses did not include a baseline period before the COVID-19 infection that might have provided more precise evaluation of baseline differences in risk between COVID-19 cases and controls.

Preexisting DM is also associated with more severe illness from COVID-19 [32] but some studies suggest that COVID-19 may be associated with new-onset diabetes. A systematic review of 8 reports of patients hospitalised in the early stages of the COVID-19 pandemic found that 14.4% of patients developed new-onset diabetes following the illness [33]. A possible effect of SARS-CoV-2 infection on pancreatic function is suggested by the finding that the virus infects pancreatic beta cells [34], reduces insulin production, and promotes beta-cell apoptosis [35]. COVID-19 may also lead to reduced physical activity and deconditioning [36] leading to greater insulin resistance. Contacts with medical care may also lead to increased opportunities to detect previously undiagnosed diabetes. Previous studies have often reported

hospital-based cohorts, with smaller samples or shorter durations of follow-up. This large population-based study shows that patients diagnosed with COVID-19 have a slightly higher baseline risk of DM. There is evidence that new-onset diabetes is increased during the acute COVID-19 illness, and there is evidence that a net increase in diabetes persists for at least 12 weeks following the COVID-19 illness before declining. There was evidence that patients diagnosed during the acute illness might be more likely to be prescribed insulin but there was no evidence for increased type I diabetes overall, nor was there evidence that glucocorticoid exposure might account for a high proportion of cases. While this study took a population perspective, clinical research is needed to determine whether subgroups of patients at greater risk of hyperglycaemia, including those with pre-diabetes, pregnancy, or polycystic ovary syndrome, are vulnerable to post-COVID-19 DM.

## Strengths and limitations

This study drew on a large, longitudinal population-based data resource that enabled us to conduct a matched analysis of mortality and new CVD and DM diagnoses for up to 1 year following COVID-19 [28]. The study drew on clinical records with several limitations. COVID-19 participants might predominantly include symptomatic cases as well as both symptomatic and asymptomatic contacts of cases who gave positive test results. We included patients with both confirmed and suspected COVID-19, but a sensitivity analysis found that restricting the analysis to PCR confirmed infections would not alter conclusions. PCR testing was associated with patient characteristics and reliance on PCR confirmation for participant selection might lead to bias. There might be a risk that the control group might be contaminated by undiagnosed or asymptomatic COVID-19 infections, this might be particularly so for the early stage of the pandemic when testing was not widely available. The Office for National Statistics Coronavirus (COVID-19) Infection Survey showed that at the height of the first wave of infection from 27 April and 10 May 2020, an average of 0.27% (95% CI: 0.17% to 0.41%) of the general population had COVID-19 [37]. From October 2020 to October 2021, fewer than 2% of the population was infected with COVID-19 in any given week. Thus, the probability of a control participant having undiagnosed COVID-19 around the index date of a matched case is less than 0.02. The database enabled adjustment for a range of important covariates, but these were not always completely recorded. In health records, values are commonly missing "not at random" making the application of imputation methods more difficult. Covariate values may change over time without being recorded at family practices. Estimates without covariate adjustment were of very similar magnitude to adjusted estimates, suggesting that there was no appreciable bias from missing or misclassified covariate values. We did not include a measure of deprivation, which is associated both with diabetes and CVD, but cases and controls were matched for family practice, providing partial control for area-based measures including deprivation. Analyses did not include measures of the severity of illness in COVID-19. However, the concept of "severity" might be difficult to operationalise in a study of COVID-19 complications because a greater number of complications might be indicative of more severe illness. We found that COVID-19 participants had higher rates of family practice consultations, consistent with their recent illness. This increased medical surveillance could be associated with more frequent opportunities for diabetes to be diagnosed. However, illness from diabetes could itself lead to increased consultations. We did not have access to data for secondary care prescribing, consequently exposure to glucocorticoids might be underestimated. We also did not have sufficient data for alcohol use, diet, physical activity, and lipid parameters that might have contributed as confounders of observed associations. Patients diagnosed with COVID-19 were generally less healthy than controls and this might, in part, account for differences in

cardiometabolic outcomes. We employed a difference-in-difference analysis that allowed for baseline differences in risk, but there might have been residual confounding if there was misclassification of risk status from incomplete data recording. The observational nature of the study limits causal inferences concerning whether the increased risk of new CVD and DM diagnoses may result from COVID-19, or whether undiagnosed CVD and DM were more prevalent among COVID-19 cases, or whether COVID-19 aggravated or altered the natural history of preexisting disease. We also acknowledge that several variants of SARS-CoV-2 were dominant at different times during the course of the study (the original wild-type virus was superseded by the Alpha variant from December 2020 and the Delta variant from June 2021 [38]), and these variants might be associated with varying biomedical and clinical outcomes. The study employed matching but we also adjusted for matching variables in order to avoid bias [39].

## Conclusions

This study provides evidence that DM incidence remains elevated for at least 12 weeks following COVID-19 infection before declining. Advice to patients recovering from COVID-19 should include measures to reduce diabetes risk, including diet, weight management, and physical activity levels, especially in view of heightened baseline risk. CVD is increased early after COVID-19 infection, mainly from pulmonary embolism, atrial arrhythmias, and venous thromboses, and these risks are increased for up to 3 months. However, people without preexisting CVD or DM who suffer from COVID-19 do not appear to have a long-term increase in incidence of these conditions.

The views expressed are those of the authors and not necessarily those of the NHS, the NIHR, or the Department of Health.

## Supporting information

**S1 Checklist. STROBE checklist.**
(DOC)

**S1 Protocol. Study protocol.**
(PDF)

**S1 Table. Medical and product codes.**
(XLSX)

**S1 Text. Flowchart showing participant selection.**
(DOCX)

**S2 Text. Code for regression model.**
(DOCX)

**S3 Text. Medical codes associated with COVID-19 index date in 428,650 COVID-19 cases.**
(DOCX)

**S4 Text. Results of Poisson regression models showing main effects of group and time period and group by time interaction.** CI, confidence interval; RR, adjusted incidence rate ratio. (Estimates were adjusted for age, ethnicity, smoking, BMI category, SBP category, Charlson score, index month and matched set.)
(DOCX)

**S5 Text. Data for Fig 3.** Adjusted rate ratios (95% confidence intervals) by 4-week period following COVID-19. (Estimates were adjusted for age, ethnicity, smoking, body mass index

category, systolic blood pressure category, Charlson score, index month, and matched set.)
(DOCX)

**S6 Text. Characteristics of case and control patients diagnosed with DM or CVD during follow-up.**
(DOCX)

**S7 Text. Results of a sensitivity analysis on 243,716 COVID-19 cases confirmed by polymerase chain reaction (PCR) test and matched controls.** Figures are frequencies except where indicated.
(DOCX)

**S8 Text. Sensitivity analysis for incident diabetes mellitus after COVID-19 adjusting for consultation frequency.** Figures are coefficients (standard errors).
(DOCX)

## Author Contributions

**Conceptualization:** Emma Rezel-Potts, Phillip J. Chowienczyk, Ajay M. Shah, Martin C. Gulliford.

**Formal analysis:** Emma Rezel-Potts, Martin C. Gulliford.

**Funding acquisition:** Emma Rezel-Potts.

**Methodology:** Abdel Douiri, Xiaohui Sun, Phillip J. Chowienczyk, Ajay M. Shah.

**Project administration:** Emma Rezel-Potts, Martin C. Gulliford.

**Supervision:** Abdel Douiri, Xiaohui Sun, Phillip J. Chowienczyk, Ajay M. Shah.

**Writing – original draft:** Emma Rezel-Potts.

**Writing – review & editing:** Abdel Douiri, Xiaohui Sun, Phillip J. Chowienczyk, Ajay M. Shah, Martin C. Gulliford.

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
