## [Editor Report · Decision Letter 0]

18 Jan 2022

Dear Dr Gulliford, 

Thank you for submitting your manuscript entitled "Differential long-term impact of Covid-19 on incidence of diabetes mellitus and cardiovascular diseases: population-based cohort study" for consideration by PLOS Medicine.

Your manuscript has now been evaluated by the PLOS Medicine editorial staff and I am writing to let you know that we would like to send your submission out for external peer review.

Please re-submit your manuscript within two working days, i.e. by Jan 20 2022 11:59PM.

Kind regards,

Callam Davidson

Associate Editor

PLOS Medicine

---

## [Decision Letter · Decision Letter 1]

17 Feb 2022

Dear Dr. Gulliford,

Thank you very much for submitting your manuscript "Differential long-term impact of Covid-19 on incidence of diabetes mellitus and cardiovascular diseases: population-based cohort study" (PMEDICINE-D-22-00177R1) for consideration at PLOS Medicine. 

Your paper was evaluated by an associate editor and discussed among all the editors here. It was also discussed with an academic editor with relevant expertise, and sent to independent reviewers, including a statistical reviewer. The reviews are appended at the bottom of this email and any accompanying reviewer attachments can be seen via the link below:

[LINK]

In light of these reviews, I am afraid that we will not be able to accept the manuscript for publication in the journal in its current form, but we would like to consider a revised version that addresses the reviewers' and editors' comments. We cannot make any decision about publication until we have seen the revised manuscript and your response, and we plan to seek re-review by one or more of the reviewers. 

We hope to receive your revised manuscript by Mar 10 2022 11:59PM. Please email us (plosmedicine@plos.org) if you have any questions or concerns.

We look forward to receiving your revised manuscript. 

Sincerely,

Callam Davidson, 

PLOS Medicine

plosmedicine.org

Please add continuous line numbering throughout the manuscript to facilitate future reviews.

Please update your title to ‘Cardiometabolic health outcomes after COVID-19 infection in the UK: A matched cohort study’, or similar.

The Data Availability Statement (DAS) requires revision. Note that a study author cannot be the contact person for the data.

Please structure your abstract using the PLOS Medicine headings (Background, Methods and Findings, Conclusions).

Abstract Background: Provide the context of why the study is important. The final sentence should clearly state the study question.

In the last sentence of the Abstract Methods and Findings section, please describe the main limitation(s) of the study's methodology.

Citations should be in square brackets, and preceding punctuation.

Please ensure that the study is reported according to the STROBE guideline, and include the completed STROBE checklist as Supporting Information. Please add the following statement, or similar, to the Methods: "This study is reported as per the Strengthening the Reporting of Observational Studies in Epidemiology (STROBE) guideline (S1 Checklist)."

Please cite your protocol in the Methods section rather than the Acknowledgements. 

Did your study have a prospective analysis plan? Please state this (either way) early in the Methods section.

The terms gender and sex are not interchangeable (as discussed in http://www.who.int/gender/whatisgender/en/ ); please use the appropriate term.

Please indicate in the figure caption the meaning of the bars in Figure 1.

Please refrain from including interpretation in the Results section (e.g. ‘The increase in CVD diagnoses was apparent in the last estimate for the pre-index period, which might imply that Covid-19 associated CVD might be recorded before Covid-19 diagnoses’ in paragraph 2 of the Results). 

Please provide titles and legends for all figures (including those in Supporting Information files).

Please define any abbreviations used in Tables/Figures in the relevant legend.

Please present the unadjusted incident rate ratios as well as the adjusted ratios in Table 3.

Please relocate the Ethical Approval information to the Methods section.

The Funding, Competing interests, Contributors, and Data sharing sections should be removed from the manuscript and captured as part of your submission form responses (in the event of publication, this information will be included as metadata).

Please ensure references are formatted per our guidelines (e.g. use of et al. only after listing the first six authors, abbreviations per those found in the National Center for Biotechnology Information (NCBI) databases). Guidelines are available at https://journals.plos.org/plosmedicine/s/submission-guidelines#loc-references

Please include [preprint] in reference 25. 

Comments from the reviewers:

Reviewer #1: It is essential to know the long-term cardiometabolic effects of COVID-19 so that measures can be taken early to prevent these after the acute phase of illness. In this context, this study's findings have important clinical implications. 

1. What type of SARS-CoV-2 variant was prevalent during the study period? 

2. Alcohol use, diet, physical activity, and lipid parameters (important risk factors for both CVD and diabetes) were not adjusted in the analysis. Are data not available? If yes, then acknowledge this as a limitation. 

3. Cases and controls were matched on age, gender, and family practice, but the analyses were adjusted for age and gender. Isn't this over adjusting?

4. Apologies if I missed this - what % of patients were on glucose-lowering drugs, and what % were on insulin in the post COVID phase?

5. Comment on the % of participants with missing covariates. 

Reviewer #2: In this study the authors examine acute, post-acute and long-term consequences of COVID-19 infection focusing primarily on cardiometabolic and pulmonary endpoints. They use cohort data from 1,473 family practices in UK and compare individuals with positive COVID-19 PCR test results to those who do not have a positive PCR test. For the analysis they remove prevalent cases of cardiometabolic diseases and examine incidence of cardiometabolic diseases after COVID-19. The study is well executed, and statistical methods are sound. The authors describe some of the limitations but there are a few additional limitations that I will address below that the authors should address.

1. The matched control individuals are healthier than those that get COVID-19. Therefore, some of the higher incidence of DM after COVID-19 may be related to higher baseline risk for DM. Please address this in the discussion.

2. Selection bias due to infection, severity of infection or availability for testing should be discussed. 

3. The authors have information on BMI and smoking only in a subset of the cohort (Table 1). To account for missingness they add a new variable for missing data. It is possible that missing data is masking a worse cardiometabolic health (like higher BMI), or increased smoking. The authors should perform sensitivity analysis in a sample without any missing data.

Reviewer #3: Thanks for the opportunity to review your manuscript. My role is as a statistical reviewer so my comments focus on the design, data and analysis presented in the manuscript. I have put general questions first, and followed these with queries relevant to a specific section of the manuscript. 

This study estimates whether people from England during 2021 were more likely to have a new diagnosis of DM or CVD in the 12 months after they were exposed to COVID-19. The data is sourced from routinely collected health data (CPRD). A cohort was created by matching patients with COVID-19 to unaffected patients, matching on age, gender, and home healthcare clinic. 

Overall I think the study design and analysis is appropriate and the manuscript is clearly written. There are some additional details that need to be added to fully understand the limitations of the data sources used in the study. I was also not clear how missing covariate data was dealt with in the analyses.

There are some differences from the planned analyses (in the protocol on CPRD) - in particular the use of Poisson regression instead of Cox models. What was the rationale for the change?

One covariate that I would consider important to consider as a confounder between COVID-19 and CVD/DM is healthcare utilisation. My experience is that DM diagnosis can be sensitive to increased use of medical care, i.e. more ill patients seek care and fasting glucose is commonly part of the standard panel of blood tests (and subsequent A1C testing). Are measures of healthcare utilisation able to be derived from the CPRD data? These could explain the more modest IRRs for DM in the post-acute and long phases. This is briefly acknowledged in the discussion, but some way of checking this would be helpful to put these findings into context. 

P2, Outcomes. Given that half of the sample were not exposed to Covid-19, i

P4, Paragraph 2. Have there been any studies with longer-term follow-up of these type of samples? 

P5, Paragraph 1. When establishing an exposure-outcome association, representativeness is not strictly necessary, but it would helpful to point to a reference or table that shows this.

Does CPRD Aurum uniformly capture patients across England or are there regions that predominate? Not necessarily an issue - but I have seen previously studies with CPRD extracts that are more regionally focused (i.e. Northern England). 

P5, Paragraph 2. How were the COVID-19 diagnoses linked to the CPRD data? Is this a routine linkage to patients from a notifiable diseases registry? 

During this stage in England, were most COVID-19 diagnoses from PCR testing? 

So just to clarify my understanding, effectively a pool of controls was created for each index date, including those not previously matched to a COVID-19 replacement (i.e. matched without replacement) and with no COVID-19 diagnosis before or on the index date? And stratified by home family practice? 

Can the exclusions of patients with the numbers (e.g. excluding those with prevalent CVD/DM) be included in a flow diagram? 

P5, Paragraph 3. From what sources do the CVD diagnoses come from? Patient self-report or linked hospitalisations? 

Can you include a supplementary list (or reference) the drugs used to classify DM status? (i.e. ATC codes etc.). Were you able to separate the drug use by indication - e.g identify where these were for Gestational DM, or metformin use for PCOS? 

P6, Paragraph 2. Generally how up to date were these covariates? 

Strictly speaking, the outcome in the study is CVD or DM, so the patients were matched on exposure status not on outcome status. I would consider adjusting the terminology (e.g. 'case and control patients') as this does give the impression that this is a case-control study with COVID-19 as the outcome. 

What was the rationale of examining glucocorticoids?

P6, Paragraph 7 (and next page). The first two sentences of this paragraph (periods of 28 days, and calculations performed in days) seem to contradict each other. Or are they referring to separate parts of the analysis?

Were the covariates in the analysis in the same form as in Table 1? Was age included as linear predictor? Was the fit of this checked?

What checks were made of the Poisson regression models - i.e. over-dispersion, distribution of residuals? 

How was missing data (i.e. covariates) managed in the analyses? 

Was the matched set the identity of each individual over the three time points? Or the pair of patients matched on age/gender/health centre? 

Supp Table 2.

How was the overall difference estimated? Was this weighted according to follow-up time in each phase of the study? 

Reviewer #4: This study aimed to quantify the risk of incident diabetes mellitus and cardiovascular disease in people registered in the CPRD Aurum primary care database as having had a positive test result for SARS-CoV-2 infection, compared to people who had no record of infection in the same data source.

The relevance of the research question is unquestionable, and the results of this study (if correct) would be very important for public health. However, I do have concerns about the suitability of the methods to address the research question. I list my main concerns below: 

1. There is a high risk of information bias due to misclassification of the exposure (infection with SARS-CoV-2), particularly in the first wave. Access to testing was practically non-existent during the first months of the pandemic in the UK. Therefore, what is being mostly captured in electronic health records from primary care is symptomatic infection, rather than all infections with SARS-CoV-2. I acknowledge that the inclusion of 'suspected' cases in the analysis might have mitigated this effect to some extent, but the potential for controls to have been positive for SARS-CoV-2 during the first wave is non-negligible. 

2. The authors state that they conducted a matched cohort study, but the results included two components:

2.1. In the first component, the authors describe the incidence of outcomes prior to the index date in the exposed and unexposed cohorts. There is vast evidence that people with comorbidities have higher risk of severe COVID-19 outcomes, should they get infected. As people with comorbidities are more likely to have more severe disease, they will also be more likely to present in primary care with symptoms and have the disease recorded in their health record. For this reason, I strongly disagree with the authors' claim in the discussion: "The study showed that patients recorded with Covid-19 in primary care are at slightly greater risk of CVD and DM even before the Covid-19 diagnosis." This study showed that people with history of CVD and DM are more likely to present with symptomatic infection of SARS-CoV-2, which we already knew. In addition, there is also a survival bias inherent in this analysis: the authors described the incidence of CVD and diabetes in two cohorts of people who survived long enough to present with symptomatic COVID. People who died from CVD are not included in the denominator.

2.2. I am also concerned with the execution of the second component of the study (i.e. the association between COVID-19 and incident CVD and DM). The authors stated that: "Matched sets that included case or control patients with prevalent CVD or DM, diagnosed within 12 months of the start of record or more than 12 months before the index date, were excluded." This is problematic. Patients with prevalent disease should have been excluded before matching, to avoid excluding pairs where the chosen control had CVD/DM history, as another control could have been selected instead of dropping the case. Also, the precise definition of the outcomes is not given (perhaps these could be included in appendix?) and thus is not clear to me what counted as an 'incident' event. For example, people with a DM diagnosis 5 years prior to COVID - were they excluded from the denominator in the descriptive 'incidence' analysis and dropped from the matched cohort study? This is unclear.

3. Following from the previous comment, could the authors provide the precise definitions of all variables used in this study and a flowchart for the patients' inclusion/exclusion. Some variables included in the models are prone to missingness but there is no mention to missing data in the methods. In line with best practice in the field, and to ensure transparency and reproducibility, could the authors make all code lists and patient selection information available in appendix? 

4. The authors used 'long-COVID' to refer to patients who had SARS-CoV-2 infection 13 to 52 weeks after the index date. I acknowledge that this is clearly defined in the paper but 'long-COVID' has been used since the beginning of the pandemic to describe the state where symptoms linger after SARS-CoV-2 infection. I kept having to remind myself that 'long-COVID' in this paper was defined by time since index. Wondered if simply listing the time since index throughout the paper would make the manuscript clearer. 

I have other minor comments: 

Abstract

The abstract needs revision as it contains several typos. COVID-19 and SARS-CoV-2 infection are used interchangeably in the abstract - could this be corrected as the two are not equivalent. As mentioned above, I would also avoid the term 'Long covid'. The section for 'Outcomes' also includes the methods description - perhaps a mistake?

Introduction

The introduction gives a good rationale for this study. However, some sentences could be revised to improve the coherency of the argument being made: 

(1) "Acute Covid-19 infection has been associated with new-onset cardiovascular disease (CVD) and diabetes mellitus (DM.)" Could you add a reference for this claim? This sounds like a fact and one wonders about what this study adds to the current body of research.

(2) "New onset hyperglycaemia has been reported in Covid-19 patients and is associated with worse prognosis" Please add a reference.

(3) "There were in excess of 1,000 deaths per day in the UK during April 2020 and January 2021" This is incorrect. According to the reference given by the authors, between April 2020 and January 2021, there were fewer than 1000 deaths registered in most of the days.

(4) "However, susceptibility to and severity of Covid-19 infection is known to be associated with cardiometabolic risk." What does susceptibility mean in this sentence? Susceptibility to the infection? 

(5) "We employed the Clinical Practice Research Datalink (CPRD), a national database of anonymised primary care electronic health records, to identify a cohort of Covid-19 patients, comparing new CVD and DM diagnoses to a matched cohort with no Covid-19 diagnosis." Please note that none of the primary care databases at CPRD has national UK coverage. CPRD GOLD is no longer representative of the UK population in terms of geographical distribution of contributing practices. CPRD Aurum includes data from GP practices in England and Northern Ireland, and do not include practices from the devolved nations. 

Methods

1. The number of practices varies across builds of the CPRD Aurum primary care database. Please clarify which build was used before stating the number of practices included in the database. 

2. Please cross-check the number of patients registered in practices actively contributing with data to CPRD Aurum in the build used for this study. The % of the population covered is probably outdate too. I suggest that the author look for these numbers in the CPRD Release notes for the specific build used in this study.

3. Minor: please delete '18.5' prior to 'Kg/m2' in all categories.

4. Could the authors also include a DAG or some justification as to why they considered those variables as potential confounders of the association between COVID-19 and incident CVD / DM outcomes? 

5. The information on PCR testing is interesting. Could the authors clarify how did they ascertain who got a PCR positive test?

6. What were mortality data used for?

[LINK]

---

## [Decision Letter · Decision Letter 2]

1 Apr 2022

Dear Dr. Gulliford,

Thank you very much for submitting your revised manuscript "Cardiometabolic outcomes up to 12 months after COVID-19 infection. A matched cohort study in the UK" (PMEDICINE-D-22-00177R2) for consideration at PLOS Medicine. 

Your paper was evaluated by an associate editor and discussed among all the editors here. It was also sent back to independent reviewers, including a statistical reviewer. The reviews are appended at the bottom of this email and any accompanying reviewer attachments can be seen via the link below:

[LINK]

In light of the reviewers' comments, we will not be able to accept the manuscript for publication in the journal in its current form, but we would like to consider a revised version that addresses the reviewers' and editors' comments. Obviously we cannot make any decision about publication until we have seen the revised manuscript and your response, and we plan to seek re-review by one or more of the reviewers. 

We hope to receive your revised manuscript by Apr 29 2022 11:59PM. Please email us (plosmedicine@plos.org) if you have any questions or concerns.

We look forward to receiving your revised manuscript. 

Sincerely,

Callam Davidson, 

PLOS Medicine

plosmedicine.org

Please provide an explanation as to why your use of the same data system is different from Rev #4.

Comments from the reviewers:

Reviewer #1: Thank you authors for addressing my comments and revising the manuscript accordingly. Please change "moderators" to "confounders" in your response to comment 2. 

Reviewer #2: The authors have sufficiently addressed all my comments.

Reviewer #3: Thanks for the revised manuscript and response to my original queries. Overall the revisions and responses clarified all of my original review and apart from one very small change I recommend that the manuscript be accepted for publication. The editor can safely ignore the cut-off comment - this should have been removed before submitting the initial review.

The explanation for the change in analysis to allow for a difference-in-difference analysis makes sense to me, and I agree that generally you will see similarity between HRs and IRRs when each analysis is applied to the same data. The DID approach with this type of data really looks effective. 

I agree with your reply to Reviewer 1, point 2 about the adjustment for age and sex. This is an approach often recommended as being 'doubly-robust' in that if one (e.g. matching) fails then the other (e.g. covariate) will be effective. Even if data is already well-matched and confounding with the exposure is 'eliminated', it is still a good idea if there is an association between these covariates and the outcome, as it shrinks the unexplained variance and you end up with more precise estimates of standard error (and hence narrower 95% CI) of other estimates in the model.

The one small pedantic change requested - for categories of BMI (L114), I would use "18.5 - <25 kg/m2", as if it were "18.5-24.9" strictly speaking someone with a BMI of 24.95 would not be classified into any category.

Reviewer #4: RE Comment 2.2: 

Thank you for clarifying what was done. The authors stated that "CVD and DM status could only be determined after controls were selected and their full data was extracted from the database". This is untrue. The 'define' tool, in the CPRD data platform, allows researchers to obtain a list of all patients with a record of a given condition (as defined in the code list). This includes CVD and DM. Once you ran the define and obtain a list of patients with history of CVD and DM, this can be merged with the denominator files (the complete list of patients that you mentioned) to identify and exclude those that are not eligible for the study. Matching should only be done after creating the pool of potentially eligible controls.

I understand this is not what was done, and I acknowledge that the authors were transparent about this in the revised version of the manuscript. I do not know if the non-standard design of the study introduced bias, and if yes, of what magnitude. I suspect this won't change the study conclusion, but I do struggle with this error that could be easily avoided. A statistical reviewer may be better suited to advise. 

Thank you for clarifying all other points and thank you for the opportunity to review this paper.

[LINK]

---

## [Decision Letter · Decision Letter 3]

18 May 2022

Dear Dr. Gulliford,

Thank you very much for submitting your revised manuscript "Cardiometabolic outcomes up to 12 months after COVID-19 infection. A matched cohort study in the UK" (PMEDICINE-D-22-00177R3) for consideration at PLOS Medicine. 

Your paper was re-evaluated by an associate editor and discussed among all the editors here. It was also discussed again with an academic editor with relevant expertise, and sent back to independent reviewers, including a statistical reviewer. The reviews are appended at the bottom of this email and any accompanying reviewer attachments can be seen via the link below:

[LINK]

In light of these reviews, I am afraid that we are still unable to accept the manuscript for publication in the journal in its current form, but we would like to consider a revised version that addresses the reviewers' and editors' comments. 

We hope to receive your revised manuscript by Jun 01 2022 11:59PM. Please email us (plosmedicine@plos.org) if you have any questions or concerns.

We look forward to receiving your revised manuscript. 

Sincerely,

Callam Davidson, 

Associate Editor

PLOS Medicine

plosmedicine.org

Please clarify your methodology in response to the comments of reviewer #4. 

Please use track changes rather than highlighting to indicate changes in your marked up manuscript – the previous version provided did not highlight all the changes made between the R2 and R3 versions.

Please ensure all figures can be interpreted without the need to refer to the main text (Figure 1 for example needs either a key or information in the legend to define which line represents patients vs controls).

Please add the following statement, or similar, to the Methods: ""This study is reported as per the Strengthening the Reporting of Observational Studies in Epidemiology (STROBE) guideline (S1 Checklist)."

Changes in the analysis-- including those made in response to peer review comments-- should be identified as such in the Methods section of the paper, with rationale.

Please ensure all supplementary tables are cited in the main text (I couldn’t locate a reference to Supplementary Table 2). 

Comments from the reviewers:

Reviewer #3: I noted after the last review that excluding the matched sets for a members with prevalent DM/CVD shouldn't introduce bias, it's reassuring to see that when this step is applied earlier there is no appreciable difference in the pattern of DM incidence. 

 The additional analysis by 4-week period with the extended data is quite interesting - better not to speculate about mechanisms, but I will just add that I think this is a valuable addition. 

One very small check, and an addition as well. For fig 3 the confidence intervals come from the SE estimates of the main model? If so then I would just then specify the method used to create the smoothed line. 

Great paper - whoever was doing the analysis might need a day off after managing to update the paper with additional data so quickly

Reviewer #4: Thank you for addressing my comments. Unfortunately, I don't think the methods are clear. The authors described using the Feb 2021 version of Aurum to identify patients with COVID-19. They then mention that controls (I am assuming they meant to say controls when they said 'patients') 'were randomly selected from the registered population of CPRD Aurum March 2021 release' (p. 7, lines 88-89). Subsequently, the authors mentioned that 'data were updated to the March 2022 release of Aurum for final analysis'. I am unclear why three different versions of CPRD Aurum were used, when all information you need is provided in the most recent version of the database. This gives me no confidence that the authors handled the data properly, and therefore I cannot recommend this paper for publication.

[LINK]

---

## [Decision Letter · Decision Letter 4]

10 Jun 2022

Dear Dr. Gulliford,

Thank you very much for re-submitting your manuscript "Cardiometabolic outcomes up to 12 months after COVID-19 infection. A matched cohort study in the UK" (PMEDICINE-D-22-00177R4) for review by PLOS Medicine.

I have discussed the paper with my colleagues and the academic editor and it was also seen again by one reviewer. I am pleased to say that provided the remaining editorial and production issues are dealt with we are planning to accept the paper for publication in the journal.

[LINK]

We hope to receive your revised manuscript by EOB Wednesday 15th June. Please email us (plosmedicine@plos.org) if you have any questions or concerns.

We look forward to receiving the revised manuscript by Jun 15 2022 11:59PM.   

Sincerely,

Callam Davidson, 

Associate Editor 

PLOS Medicine

plosmedicine.org

Requests from Editors:

In the last subsection of the abstract, please update to “In this study, we found that CVD was increased early after Covid-19 mainly from pulmonary embolism, atrial arrhythmias and venous thromboses. DM incidence remained elevated for at least 12 weeks following Covid-19 before declining.”

Please ensure journal abbreviations in your references align with those found in the National Center for Biotechnology Information (NCBI) databases. 

Comments from Reviewers:

Reviewer #4: Thank you for addressing my comments. The revised version of the manuscript is much clearer in the description of the methods.

[LINK]

---

## [Editor Report · Decision Letter 5]

14 Jun 2022

Dear Dr Gulliford, 

On behalf of my colleagues and the Academic Editor, Dr Weiping Jia, I am pleased to inform you that we have agreed to publish your manuscript "Cardiometabolic outcomes up to 12 months after COVID-19 infection. A matched cohort study in the UK" (PMEDICINE-D-22-00177R5) in PLOS Medicine.

When making the formatting changes, please also address the following comments:

* Apologies that my previous comment was unclear - please re-insert the final sentence from the previous draft of your manuscript into the Abstract Conclusions ('People without pre-existing CVD or DM who suffer from COVID-19 do not appear to have a long-term increase in incidence of these conditions.'). 

* Please remove 'Funding: UK National Institute for Health Research.' from the Abstract.

PRESS

Sincerely, 

Callam Davidson 

Associate Editor 

PLOS Medicine